# Observation of quadratic Weyl points and double-helicoid arcs

Hailong He [1], Chunyin Qiu [1✉], Xiangxi Cai[1], Meng Xiao [1], Manzhu Ke [1], Fan Zhang [2] & Zhengyou Liu [1,3✉]

Novel quasiparticles beyond those mimicking the elementary high-energy particles such as Dirac and Weyl fermions have attracted great interest in condensed-matter physics and materials science. Here we report an experimental observation of the long-desired quadratic Weyl points by using a three-dimensional chiral metacrystal of sound waves. Markedly different from the newly observed unconventional quasiparticles, such as the spin-1 Weyl points and the charge-2 Dirac points featuring respectively threefold and fourfold band crossings, the charge-2 Weyl points identified here are simply twofold degenerate, and the dispersions around them are quadratic in two directions and linear in the third one. Besides the essential nonlinear bulk dispersions, we further unveil the exotic double-helicoid surface arcs that emanate from a projected quadratic Weyl point and terminate at two projected conventional Weyl points. This unique global surface connectivity provides conclusive evidence for the double topological charges of such unconventional topological nodes.

[1] Key Laboratory of Artificial Micro- and Nano-Structures of Ministry of Education and School of Physics and Technology, Wuhan University, Wuhan 430072, China. [2] Department of Physics, University of Texas at Dallas, Richardson, TX 75080, USA. [3] Institute for Advanced Studies, Wuhan University, Wuhan 430072, China. ✉email: cyqiu@whu.edu.cn; zyliu@whu.edu.cn

Discovering new topological matter has been one of the key themes in fundamental physics and materials science[1,2]. Recently, great efforts have been devoted to the exploration of topological semimetals[3] that are featured with nontrivial band crossings at Fermi energy. Prime examples include Dirac and Weyl semimetals[4–7], which host isolated fourfold and twofold linearly degenerate Fermi points, dubbed Dirac, and Weyl points, in their three-dimensional (3D) Brillouin zones (BZ). These linear band crossings give rise to the condensed-matter counterparts to the elementary Dirac and Weyl particles in high-energy physics. Remarkably, the Weyl points act as the monopoles of Berry curvature and carry quantized topological charges (the first Chern numbers) of ±1, enabling many exotic properties, such as the long-thought bulk chiral anomaly[3] and the unprecedented surface Fermi arcs[6–8].

Most recently, novel quasiparticles beyond the conventional Dirac and Weyl ones, without any high-energy counterparts, have attracted tremendous attention[9–17]. Such unconventional quasiparticles, protected by crystalline symmetries in structurally chiral crystals, correspond to multifold (i.e., more than twofold) band crossings and carry higher topological charges. Among them, the so-called spin-1 Weyl points and charge-2 Dirac points have been experimentally realized[13–17] in solid-state materials soon after the theoretical predictions[9–12]. Unlike the conventional Weyl points (CWPs), the spin-1 Weyl points are crossed with a linear Weyl cone and one extra flat band, whereas the charge-2 Dirac points are formed with two CWPs of the same monopole charges. Both quasiparticles carry topological charges of ±2, which can emerge at different time-reversal-invariant momenta of the same system. In contrast to the CWP systems (e.g., TaAs (refs. [6,7])), the symmetry and topology underlying the structural chirality enforce giant surface Fermi arcs and produce many intriguing physical properties, such as the quantized circular photogalvanic effect[18]. Besides these complex multifold degeneracies, unconventional charge-2 nodal points have also been predicted in the simplest twofold band crossings[19], around which the dispersions are linear in one direction yet quadratic in the other two. To the best of our knowledge, however, such quadratic Weyl points (QWPs) have yet to be observed in solid-state systems due to the lack of ideal material candidates, despite proposed much earlier than the aforementioned charge-2 quasiparticles.

Metacrystals, periodic artificial structures for classical waves, have been proven to be elegant platforms for creating and manipulating topological states of matter[20–23]. Particularly, the macroscopic characteristics of metacrystals enable flexible engineering of surface states and direct visualization of various exceptional surface phenomena[24–29]. Although the QWPs have been predicted in several metacrystals[24,27,30], no compelling experimental evidence for their existence has yet been offered. Here, we present the first experimental observation of this elusive, yet highly desirable, QWPs in a 3D chirally stacked acoustic metacrystal[27]. The extremely clean bulk band structure hosts both the charge-1 CWPs and charge-2 QWPs. Markedly, all these topological nodal points are pinned by symmetry to the widely separated high-symmetry momenta in the bulk BZ, which facilitates the experimental identification of the characteristic bulk nodes and surface arcs. Interestingly, this acoustic chiral crystal has been recently utilized to reveal novel topological negative refraction effect[27] at a frequency near the CWP. In this work, we focus on the topological physics around the QWP. Based on our airborne sound experiments, we observe not only the unusual quadratic dispersions around the QWPs, but also the intricate double-helicoid surface arcs emanating from the projected QWPs (ref. [31]). We further reveal that, because of the coexistence of CWPs and QWPs, the surface arcs exhibit a unique global connectivity in the frequency–momentum space. Our systematic experimental results agree excellently with our numerical simulations performed with COMSOL Multiphysics (Methods section).

## Results

**Acoustic chiral metacrystal and its bulk band structure.** Figure 1a displays the real structure of our acoustic metacrystal. It is a chiral stack of identical square acrylic rods along the z-direction; each adjacent pair is twisted anticlockwise by $2\pi/3$, forming a triangular lattice when projected to the x–y plane. The side length of the square rods is $b = 9.8$ mm, the in-plane and out-of-plane lattice constants are $a = h = 29.4$ mm, and the rest volume is filled with air. This 3D structure lacks mirror and inversion symmetries, and belongs to a nonsymmorphic space group. Figure 1b shows its band structure hosting both CWPs and QWPs (refs. [27,30]). Remarkably, all the band crossing points locate exactly at the high-symmetry momenta of the 3D BZ as seen in Fig. 1b, c: the charge-1 CWPs emerge at K (K′) and H (H′), and the charge-2 QWPs are pinned to the time-reversal-invariant momenta, Γ and A. The latter is consistent with the fact that the time-reversal symmetry relates two CWPs of the same charge. The overall Weyl node distribution is a consequence of the threefold screw symmetry of our structurally chiral metacrystal[30]. As exhibited more clearly in Fig. 1d, while the dispersions simulated around the CWPs are linear in all directions, the dispersions around the QWPs are quadratic in the $k_x$ and $k_y$ directions, though linear in the $k_z$ direction. The topological charges of the CWPs and QWPs, obtained by analyzing the eigenvalues of the threefold screw symmetry at their high-symmetry momenta[19,30], are highlighted in Fig. 1b, c. The total charge is zero, consistent with a no-go theorem. It is worth pointing out that, similar to those newly unveiled unconventional point nodes[9–17], all the Weyl nodes here are stable against pair annihilation, given their symmetry-enforced high-symmetry-point locations. This also results in long surface arcs and thus benefits their experimental detections. Moreover, as a direct result of our chiral structure, the oppositely charged QWPs or CWPs are at different frequencies, which enables a broadband for studying their local and global topological characteristics.

The presence of QWPs can be demonstrated by our airborne sound experiments. Figure 2a illustrates our experimental setup (Methods section). To excite as many bulk states as possible, a broadband point-like sound source was inserted and fixed in the middle of sample. A 1/4-inch microphone (B&K Type 4187) was moved inside the sample to probe the pressure distribution in a horizontal plane slightly above the source (Fig. 2a). Performing 2D spatial Fourier transformation of the measured pressure field, we obtain the projected band structure in the $k_x$–$k_y$ plane (see inset). Figure 2b shows the data extracted along the path $\bar{M}$–$\bar{\Gamma}$–$\bar{K}$–$\bar{R}$ in the 2D BZ. Overall, the experimental data agree excellently with the numerical simulations (color lines). In particular, two quadratic band touching points are clearly observed at $\bar{\Gamma}$: the higher-frequency one corresponds to the QWP at Γ, whereas the lower-frequency one corresponds to the QWP at A. By contrast, the projected dispersions around $\bar{K}$ exhibit in-plane linearity for the CWPs at H and K. In a similar fashion, we further obtain the projected band structure in the $k_x$–$k_z$ plane, as shown in Fig. 2c. Besides the linear dispersions around the CWPs, the linearity, though not fully captured due to the masking effect (by the projected bulk states other than those along Γ–A), is displayed clearly in the lowest (middle) band connecting the lower-frequency (higher-frequency) QWP along the $\bar{\Gamma}$–$\bar{A}$ direction.

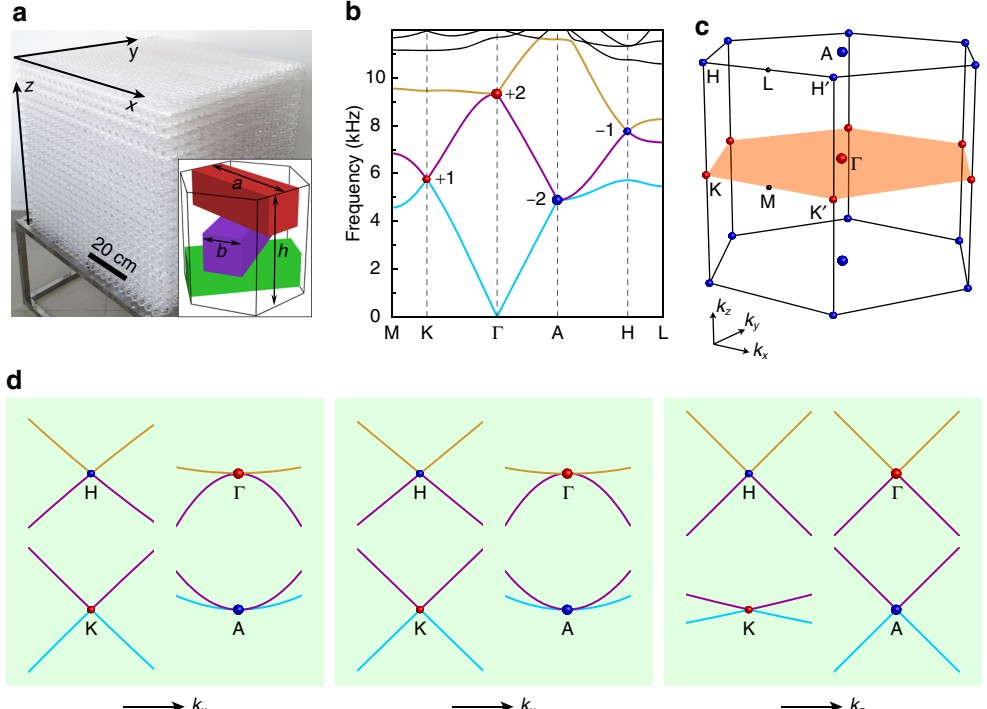

**Fig. 1 3D chirally stacked acoustic metacrystal with multiple Weyl points. a** An image of the sample and its unit-cell geometry. **b** Band structure calculated along several high-symmetry lines, where the color spheres highlight the differently charged topological nodes provided by the lowest three bands (color lines). **c** Bulk BZ of the metacrystal. **d** Local dispersions simulated around the four nodal points, revealing clearly the quadratic dispersions near the QWPs $\Gamma$ and A, along the $k_x$ and $k_y$ directions, in contrast to the CWPs K and H crossed linearly along all directions.

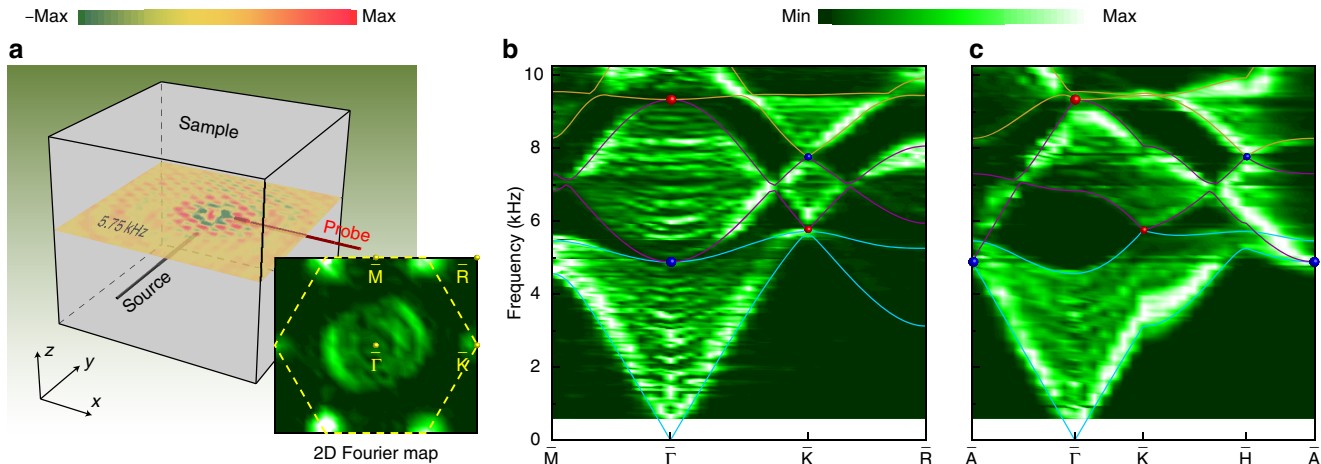

**Fig. 2 Experimental evidence for the quadratic dispersions around QWPs. a** Experimental setup. Inset: 2D Fourier transformation of the pressure field measured in a horizontal $xy$ plane inside the sample, exemplified at the frequency (5.75 kHz) of the CWP at K. The dashed lines sketch the 2D projection of the bulk BZ. **b** Projected band structure (color scale) extracted along the high-symmetry lines. The color lines sketch the boundaries of the projected lowest three bulk bands simulated with COMSOL Multiphysics, and the color spheres highlight the projected Weyl nodes. **c** Similar to **b**, but for the 2D Fourier transformation of the pressure field scanned in a vertical $xz$ plane. The corresponding 2D projection of the bulk BZ is sketched in Fig. 3a.

**Double-helicoid topological surface arcs**. As a defining feature of Weyl semimetals, the presence of topological surface arcs has been considered as smoking gun evidence for their nontrivial topological properties[3,8]. In turn, we can identify the topological charge of a bulk Weyl node by examining the surface-state chirality along a closed loop encircling the projected Weyl node. Figure 3a illustrates the essential physics in this method. Consider a tube orientated along the $k_y$ direction in the bulk BZ, on which the 2D bands carry well-defined Chern numbers. According to the bulk-edge correspondence, if the metacrystal is terminated at

an $XZ$ surface, the chirality of topological surface states along the tube-projected loop gives the Chern numbers of the associated 2D band gaps, from which one can deduce the total topological charge of the Weyl nodes inside the tube. In particular, if there is only one node inside the tube, its topological charge can be determined directly by the overall chirality of the topological surface states. This is the case for the $XZ$ surface, at which all the projected nodes can be distinguished from each other. Figure 3b shows the surface-projected dispersion along a small loop centered at $\bar{\Gamma}$. Evidently, two gapless surface states with overall

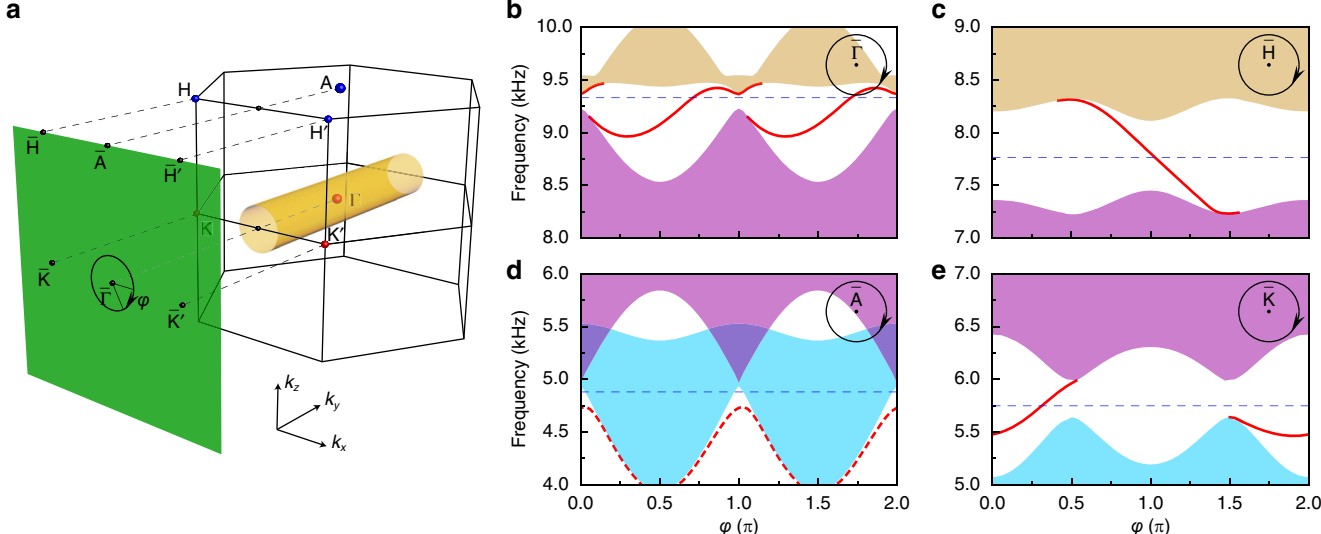

**Fig. 3 Numerical evidence for the topological charges of QWPs through nontrivial surface states. a** Bulk BZ and its surface projection to the $k_x$–$k_z$ plane. The black circle (of radius $0.2\pi/a$) is the projection of the orange tube surrounding the QWP at Γ. **b** Projected band structure along the circular path in **a**. The color shadows are the projections of different bulk bands, the red solid lines indicate the presence of topological surface states, and the blue dashed line labels the frequency of the Weyl node at Γ. **c–e** Similar to **b**, but for the paths each encircling a different Weyl node from Fig. 1b. The red dashed line in **d** corresponds to topologically trivial surface states.

positively slopes traverse the full gap between the middle and highest projected bands, indicating the +2 topological charge of the higher-frequency QWP at Γ. For comparison, we also simulated the surface-projected dispersions along the loops centered at H̄, K̄, and Ā, respectively. As expected, the projected dispersion encircling H̄ (Fig. 3c) or K̄ (Fig. 3e) features a single chiral band with a negative or positive slope inside the full gap. For the loop enclosing Ā (associated with the lower-frequency QWP at A), however, there is no gap between the projected lowest two bulk bands (Fig. 3d), consistent with the two upward-sloping bulk dispersions along A–H (Fig. 1b). As such, no topologically nontrivial surface states can be expected with bulk–boundary correspondence. (Note that the red dashed line corresponds to topologically trivial surface states since it is non-chiral along the loop). For this reason, below our experiments focus on the higher-frequency QWP at Γ. The case becomes a bit more complex for the YZ or XY surface, at which the projections of different topological nodes may overlap (see more details in Supplementary Figs. 1 and 2).

We further performed surface measurements (Methods section) to reveal the elusive double-helicoid surface arcs emanating from the projected QWP at Γ̄. Unlike the measurements of bulk states, as illustrated in Fig. 4a, an acrylic plate (blue cover) was tightly covered on the XZ surface to mimic the hard boundary in our simulations. To minimize the finite-size effect and to maximize the surface-state excitation, we launched two independent excitations at the sample corners (black stars), each igniting one surface arc according to the known group velocity (see Supplementary Fig. 3). A 1/4-inch probe was inserted between the sample and the cover from the side to detect the surface signal. We mapped out the isofrequency contours of the surface states through 2D Fourier transformation. Figure 4b shows such experimental data for a series of frequencies, descending from the frequency of the QWP at Γ to that of the CWPs at H and H′. Overall, our experimental observation matches well with the numerical simulations (red lines), despite some band broadening due to the finite-size effect. At the QWP frequency (9.33 kHz), the Fourier map exhibits clearly two surface

arcs (bright color) emanating from the charge-2 QWP. As the frequency decreases and approaches the CWP frequency (7.76 kHz), the projected QWP is masked by the projected bulk states, and the two surface arcs are respectively connected to the projected Weyl pockets surrounding H̄ and H̄′. Note that this global surface connectivity is qualitatively different from the recently discovered chiral crystals with spin-1 Weyl points and charge-2 Dirac points[13–17], in which an isofrequency contour forms an noncontractible close loop around the surface BZ torus[12,29], as illustrated in the lower panel of Fig. 4c. Here, the isofrequency contour features two connected open arcs that are time-reversal partners, and the open contour is also noncontractible because of the symmetry-enforced high-symmetry-point locations of the two inequivalent CWPs at H̄ and H̄′, as illustrated in the upper panel of Fig. 4c. In addition, the frequency evolution of the isofrequency contours exhibits double-helicoid spiraling around Γ̄ yet single-helicoid spiraling of the opposite helicity around H̄ or H̄′. The unique surface connectivity is further shown in Fig. 4d, e, the surface spectra extracted along the loops centered at Γ̄ and H̄, respectively. Here, the chosen loops are larger than those in Fig. 3 in order to display larger gaps. Consistently with the topological charges of the QWP and CWP at Γ̄ and H̄, respectively, Fig. 4d features two gapless surface states with the same chirality across the gap between the middle and highest projected bands, whereas Fig. 4e features only one gapless surface state with the opposite chirality. More discussions of the topological surface states are provided in Supplementary Figs. 4–6.

## Discussion
We have unambiguously observed and characterized the long-desired charge-2 QWPs and the consequent double-helicoid surface arcs emanating from and spiraling around the surface projection of a QWP. Both the local bulk dispersions near the QWPs and the global surface connectivity of the double-helicoid arcs to the QWPs and CWPs are unprecedented, in sharp contrast to all the other gapless topological matter

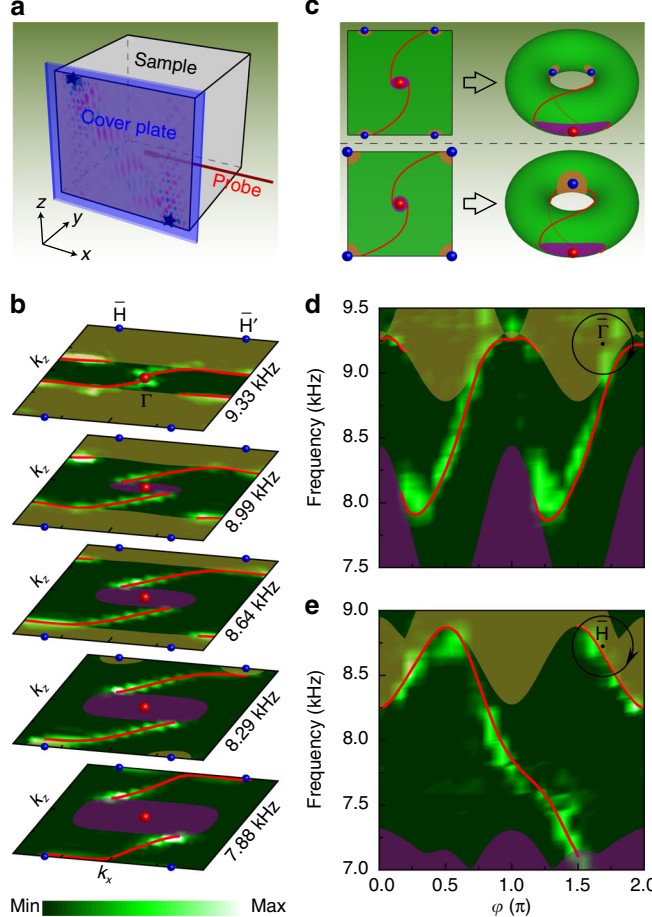

**Fig. 4 Experimental evidence for the double-helicoid surface arcs around** $\bar{\Gamma}$. **a** Experimental setup for surface measurements. **b** 2D Fourier transformations (color scale) of the measured surface fields at different frequencies, compared with the numerical isofrequency contours of the topological surface states (red lines). As before, the color spheres highlight the projected Weyl nodes, and the shaded areas are the projected bulk bands. **c** Schematic illustration of the surface arc connectivity between one charge-2 QWP and two charge-1 CWPs in our system (top), compared with that between the spin-1 Weyl point and charge-2 Dirac point in recent studies[12,29] (bottom). **d** Surface spectrum extracted along the loop of radius $0.5\pi/a$ centered at $\bar{\Gamma}$. **e** Surface spectrum extracted along the loop of radius $0.43\pi/a$ centered at $\bar{H}$.

realized to date[4–7,13–17,25,26,29]. In particular, our acoustic metacrystal is structurally simple and hosts three bands with connected QWPs and CWPs over a broad frequency range, providing an ideal platform for exploring the intriguing Weyl physics. Having provided the first experimental evidence for the elusive QWPs in acoustic systems, our results may advance the study of the unconventional quasiparticles of high topological charges. Note that the isofrequency contours of topological surface states can be flexibly tailored through engineering the sample terminations[27], which may trigger new opportunities for controlling classical waves in a topological fashion, such as realizing reflectionless interface responses to sound[26–29].

## Methods

**Simulations**. All simulations were implemented with COMSOL Multiphysics, a commercial finite-element solver. The acrylic material used for sample fabrication, which has a mass density of 1.18 g cm$^{-3}$, longitudinal velocity of 2.73 km s$^{-1}$, and transversal velocity of 1.46 km s$^{-1}$, can be modeled as acoustically rigid for airborne

sound-associated mass density 1.29 g m$^{-3}$ and sound speed 346 m s$^{-1}$, due to the huge mismatch of acoustic impedance. The bulk bands were simulated by using a single unit cell by imposing Bloch boundary conditions along all directions. The surface bands were calculated by using slab structures, with rigid boundaries in the thickness direction but Bloch boundary conditions in the remaining directions. The surface bands were distinguished from the projected bulk bands by examining the eigenfield distributions. Furthermore, we obtained the isofrequency contours of the topological surface states by scanning the entire surface BZ.

**Experiments**. The well-controlled structure design and the less demanding signal detection enabled our acoustic metacrystal to be an exceptional platform for exploring the intricate Weyl physics. Our experimental sample (Fig. 1a), prepared precisely by laser cutting technique, had $24 \times 27 \times 22$ structural periods along the $x$-, $y$-, and $z$-directions, respectively. To experimentally excite the bulk or surface states, a broadband sound signal launched from a deep subwavelength tube acted as a point-like sound source. The pressure field was scanned point-by-point through a 1/4-inch microphone (B&K Type 4187), associated with spatial steps 9.8 mm, 25.4 mm, and 29.4 mm along the x-, y-, and z-directions, respectively. Together with another identical microphone being fixed for the phase reference, both the amplitude and phase information of the pressure field were recorded and frequency resolved (at a step of 32 Hz) by a multianalyzer system (B&K Type 3560B). To map out the projected bulk bands and surface bands, 2D Fourier transformation was performed for the measured spatial pressure distributions at each given frequency, which further provided the frequency spectra along the desired momentum lines or loops. The momentum resolution of the Fourier spectrum depends on the spatial extension, where the experimental data considered. For example, $\delta k_x = \frac{2\pi}{21a}$, $\delta k_y = \frac{\pi}{10\sqrt{3}a}$, and $\delta k_z = \frac{\pi}{10a}$ were involved in Fig. 2b, c. (To reduce the boundary effects, the data near the boundaries of the measured plane were not taken into account when performing Fourier transform).

## Data availability

The data that support the plots within this paper and other findings of this study are available from the corresponding authors upon reasonable request.

## Code availability

Numerical simulations in this work are all performed using the 3D acoustic module of a commercial finite-element simulation software (COMSOL MULTIPHYSICS). All related codes can be built with the instructions in the Method section.

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

## Acknowledgements

C.Q. was supported by National Natural Science Foundation of China (grant no. 11674250) and the Young Top-notch Talent for Ten Thousand Talent Program (2019-2022). Z.L. was supported by the National Key R&D Program of China (grant no. 2018YFA0305800) and National Natural Science Foundation of China (grant nos. 11890701 and 11774275). F.Z. was supported by Army Research Office under grant no. W911NF-18-1-0416 and Natural Science Foundation under grant no. DMR-1921581 through the DMREF program. C.Q. thanks M.-L. Chang for helpful discussions.

## Author contributions

C.Q. and Z.L. conceived the idea, and supervised the project. H.H. did the simulations and experiments. C.Q., H.H., M.X., X.C., M.K., F.Z., and Z.L. analyzed the data, and wrote the manuscript. All authors contributed to scientific discussions of the manuscript.

## Competing interests

The authors declare no competing interests.
