## [Peer Review File · Nature Communications]

Reviewers' comments:

Reviewer #1 (Remarks to the Author):

In this work, the authors proposed the experimental observations of quadratic Weyl points and consequent double-helicoid surface arcs based on the woodpile acoustic metacrystal. The results are original and interesting.

I think it is worth being published in Nature Communications, but needs to be improved for better clarity according to the following points.

1. The authors should provide the acoustic parameters of the materials used in this work. Does the material itself affect the properties of the QWP?
2. Has the viscous loss been considered in the simulations? If not, what will happen after introducing the viscous loss, which exists in the real space for sure.
3. It is clear that the QWPs are quadratic in k_x - k_y plane from Fig. 2b for sure. However the linearity along k_z direction for the QWP at Γ point is not displayed very clearly in Fig. 2c, from which one cannot observe the linear dispersions like the ones in Fig. 1d. Maybe the authors should make the illustration more clear to the readers.
4. In Page 9, some more details about "finite-size effect" should be helpful to the readers.
5. The reason why the two independent point sources are used to excite sound waves in Fig. 4 is not very clear. The authors should provide more explanations.
6. When measuring the surface state in Fig. 4, a "rigid" plate was used to mimic the hard boundary in the simulation. Will it still work if it is replaced by a "soft" plate, which also prevents sound waves, and why? Will it affect the properties of surface states if there is a gap between the rigid plate and the sample?

Reviewer #2 (Remarks to the Author):

In the manuscript "Observation of a quadratic Weyl points and double-helicoid arcs", the authors present experimental evidences for charge-2 Weyl point degeneracies in a 3D acoustic metamaterial.

The systems studied by the authors has been proposed to realize double Weyl points both for acoustic and photonic crystals. In all its previous experimental realizations, however, the focus has always been on the surface physics rather than on the bulk spectrum. Therefore, the results presented are novel and of interest.

I have few remarks about the manuscript and the data analysis that I hope will be addressed in the potential resubmission:

- A low energy Hamiltonian is used to introduce the idea of multi-Weyl points. It would be nice and helpful if the parameters of this low energy would be fitted to the experimental data at a later stage in the manuscript.
- Why the author scan in a plane right above the source? Is it because of saturation issues or just because the presence of the exciter prevents to scan that very same plane?

- The inset of Fig.2 (b) and (c) are redundant given the lines in the main figure. In particular, for Fig.2(c), it covers the experimental data for the in-plane dispersion of the charge-2 Weyl point at A.
- What are the trivial surface states of Fig.3(d). Do the author understand why they appear? What is their extent in the surface Brillouin zone rather than as a function of ϕ ?
- I am slightly confused by the choice of the parameter ϕ . Currently it is taken in the counter-clockwise orientation. This gives states with a negative dispersion at γ for example. But the topological charge of γ is positive. It is purely a matter of convention but I find it confusing. I suggest to choose a clockwise convention for ϕ . Moreover, cylinders of the same size in Fig.3 and Fig. 4 would highly benefit the comparison between simulations and experimental data. Could the author choose a larger cylinder also in Fig.3?
- In Fig.4, why do the author overlay to the experimental data the simulated bulk states? I would possibly find these results more convincing if the experimentally measured bulk bands were displayed.
- Two sources are used to excite the surface states. This, however, destroys the phase information for the measured signal. How do the authors circumvent this problem? Do they Fourier transform the signal of each exciter individually and then add the incoherently (with a square-sum)?
- Which smoothing function has been applied to the experimental data?
- The authors could further elaborate on the open questions related to this work as the possible applications or possible transport studies in this platform under artificial fields.

The manuscript presents neat experimental data that substantiate the observation of a charge-2 Weyl point in an acoustic metamaterial. I believe that if the issues raised in this report will be addressed satisfactorily, the manuscript could meet the criteria of Nature Communications.

Reviewer #3 (Remarks to the Author):

In their manuscript "Observation of quadratic Weyl points and double-helicoid arcs", the authors present the first experimental realization of the quadratic Weyl points in a 3d acoustic metacrystal.

The manuscript is well written, the introduction is enjoyable to read, and the results conclusively show the properties expected. Therefore, I recommend its publication in Nature Communications once the authors address the following comments.

1. My biggest concern comes from that the acoustic structure adopted in this work is very similar to the one in their previous work titled "Topological negative refraction of surface acoustic waves in a Weyl phononic crystal", which was published in Nature. It looks like they study the same structure but focus on different properties, which reduces the novelty of this work. The authors should emphasize more standouts of it compared to their previous work.
2. In 2nd paragraph, 4th page, can the authors give the explicit expressions of coefficients A-F for their proposed structure? Is it possible to execute the calculation of Chern number (topological charge) using their effective Hamiltonians?
3. Since topological charge at A is -2, I would expect surface states present near A are topologically protected, but why the authors claim surface states encircling around Weyl node A are topologically trivial? How is the bulk-edge correspondence established in their case?
4. Some experimental details need to be supplied. For example, what is the frequency step in the frequency sweep (Fig. 2b,2c, Fig. 4d, 4e)? What are the quality factors of bulk modes and surface modes? In the experimental set up, what are the boundary conditions in z and x directions? and how are these boundary effects avoided or reduced on the Fourier spectrum?

Response to Referees

Summarized response:

We have received three reports. All the referees highly evaluated our work.

- (1) The first referee pointed out *“The results are original and interesting”* and *“it is worth being published in Nature Communications”*;
- (2) The second referee considered *“the results presented are novel and of interest”* and *“The manuscript presents neat experimental data...the manuscript could meet the criteria of Nature Communications”*;
- (3) The third referee pointed out *“The manuscript is well written, the introduction is enjoyable to read, and the results conclusively show the properties expected. Therefore, I recommend its publication in Nature Communications...”*.

Below, we provide our detailed replies to all their questions and suggestions, which are insightful and constructive to improve our manuscript. Our manuscript has been carefully revised accordingly (highlighted in yellow). Thank you very much for your reconsideration.

Reviewer #1 (Remarks to the Author):

In this work, the authors proposed the experimental observations of quadratic Weyl points and consequent double-helicoid surface arcs based on the woodpile acoustic metacrystal. The results are original and interesting. I think it is worth being published in Nature Communications, but needs to be improved for better clarity according to the following points.

Reply: We thank the referee for the high evaluation on our work and the kind recommendation for its publication in *Nature Communications*.

1. The authors should provide the acoustic parameters of the materials used in this work. Does the material itself affect the properties of the QWP?

Reply: We thank the referee for the good suggestion. Now we have added the acoustic parameters for the materials involved in this work. Actually, the properties of QWP will not be affected if the acrylic material is replaced by other commonly-used solid materials, which can be viewed as acoustically rigid for airborne sound due to the great impedance mismatch between them.

This point has been addressed in the revised manuscript (lines 218-221) “The acrylic material used for sample fabrication, which has a mass density of 1.18 g/cm³, longitudinal velocity of 2.73 km/s and transversal velocity of 1.46 km/s, can be modeled as acoustically rigid for airborne sound associated mass density 1.29 g/m³ and sound speed 346 m/s, due to the huge mismatch of acoustic impedance.”

2. Has the viscous loss been considered in the simulations? If not, what will happen after introducing the viscous loss, which exists in the real space for sure.

Reply: Our simulations, which focus on the bulk and surface band structures, ignore the viscous loss. Imaginary frequency will emerge in the dispersions if the viscous loss is taken into account, which could be negligible here since no resonance is involved in our system.

3. It is clear that the QWPs are quadratic in k_x - k_y plane from Fig. 2b for sure. However the linearity along k_z direction for the QWP at Γ point is not displayed very clearly in Fig. 2c, from which one cannot observe the linear dispersions like the ones in Fig. 1d. Maybe the authors should make the illustration more clear to the readers.

Reply: We thank the referee for pointing out this issue. Different from the bulk dispersion (Fig. 1d) calculated directly along the k_z (i.e. Γ -A) direction, where linearity is clearly illustrated near the QWP, Fig. 2c presents the surface-projected bulk bands, in which the linear dispersion could be masked by the projected bulk states other than the states along Γ -A. Now this point has been explained more carefully in our revised manuscript (lines 124-127), “Besides the linear dispersions around the CWPs, the linearity, though not fully captured due to the masking effect (by the projected bulk states other than those along Γ -A), is displayed clearly in the lowest (middle) band connecting the lower-frequency (higher-frequency) QWP along

the $\bar{\Gamma}$ - \bar{A} direction.”.

4. In Page 9, some more details about “finite-size effect” should be helpful to the readers.

Reply: We thank the referee for the thoughtful suggestion. Finite sample size will result in discretized momentum in the Fourier spectrum, in which the resolution of the momentum behaves $\sim 2\pi/L$, with L being the length of the sample. To respond this point, we have added two sentences in the method (lines 240-244), “The momentum resolution of the Fourier spectrum depends on the spatial extension where the experimental data considered. For example, $\delta k_x = \frac{2\pi}{21a}$, $\delta k_y = \frac{2\pi}{20\sqrt{3}a}$, and $\delta k_z = \frac{2\pi}{20a}$ were involved in Figs. 2b and 2c.”

5. The reason why the two independent point sources are used to excite sound waves in Fig. 4 is not very clear. The authors should provide more explanations.

Reply: There are two reasons for using such two independent point sources for excitation.

- (1) Reduce the finite-size effect. We locate the point source at the *corner* of the sample to effectively maximize the propagation distance of specific surface states, which improve the momentum resolution of Fourier spectrum.
- (2) Improve the excitation efficiency of surface states. The above treatment, instead of locating the source at the sample center, reduces the excitation efficiency of the surface states. This is improved by locating two independent point sources at different sample corners, each igniting one surface arc according to the known group velocity.

Explanations have been provided in lines 176-178, “To minimize the finite-size effect and to maximize the surface-state excitation, we launched two independent excitations at the sample corners (black stars), each igniting one surface arc according to the known group velocity.”

6. When measuring the surface state in Fig. 4, a “rigid” plate was used to mimic the hard boundary in the simulation. Will it still work if it is replaced by a “soft” plate, which also prevents sound waves, and why? Will it affect the properties of surface states if there is a gap between the rigid plate and the sample?

Figure: Projected band structures plotted along the circular path centered at $\bar{\Gamma}$, which

are simulated with rigid (left) and soft (right) boundary conditions, respectively. As we can see, there are no qualitative difference between the topological surface states (red lines) of these two cases.

Reply: Topological surface arc states will survive if the hard plate is replaced by a “soft” one (which is not easy to be realized in real experiments), since it can also be viewed as a topologically trivial insulator. Comparative numerical results can be seen in the above figure, the left for rigid boundary and the right for soft boundary. The properties of the surface states will be affected if there is an air gap between the rigid plate and the sample. Depending on the width of the air gap, the influence could be quantitative (narrow gap) or qualitative (wide gap). In particular, when the air gap becomes wider and wider, more and more trivial waveguide states will emerge and hybrid with the topological ones. A detailed discussion on this issue can be referred to *Phys. Rev. Appl.* **10**, 014017 (2018), which have been added as Ref. 28 now.

Reviewer #2 (Remarks to the Author):

In the manuscript "Observation of a quadratic Weyl points and double-helicoid arcs", the authors present experimental evidences for charge-2 Weyl point degeneracies in a 3D acoustic metamaterial. The systems studied by the authors has been proposed to realize double Weyl points both for acoustic and photonic crystals. In all its previous experimental realizations, however, the focus has always been on the surface physics rather than on the bulk spectrum. Therefore, the results presented are novel and of interest. I have few remarks about the manuscript and the data analysis that I hope will be addressed in the potential resubmission:

Reply: We thank the referee for his/her great effort made to review our manuscript. We also appreciate the referee for pointing out that "*the results presented are novel and of interest*". All the suggestions and comments below are very valuable and greatly helpful to improve our manuscript.

- A low energy Hamiltonian is used to introduce the idea of multi-Weyl points. It would be nice and helpful if the parameters of this low energy would be fitted to the experimental data at a later stage in the manuscript.

Reply: We thank the referee for the insightful suggestion. We agree with the referee that a fit of the low-energy Hamiltonian can improve the manuscript. After a careful inspection, we regrettably find that the provided Hamiltonian, which was derived straightforwardly from Refs. 19 and 30, cannot fully capture the feature of the quadratic dispersions around the DWP. Specifically, that Hamiltonian demands quadratic dispersions $|c|q^2$ and $-|c|q^2$ for the upper and lower bands touching at the DWP. However, this is not true for a general case, in which the quadratic curvatures of the two bands are independent, even hosting the same sign (e.g. the DWP at the point A, see Fig. 1d). Beyond doubt, it is very valuable to solve this problem, which is in fact what we have tried to do in the past month. However, we have not reached a satisfactory answer. Therefore, we decide to retain it as an open question and remove the effective Hamiltonian presented previously. In our opinion, this will not influence the main theme, novelty and impact of our *experimental* work.

- Why the author scan in a plane right above the source? Is it because of saturation issues or just because the presence of the exciter prevents to scan that very same plane?

Reply: The referee is correct. The presence of the exciter will prevent, to some extent, the measurement of pressure field around it.

- The inset of Fig.2 (b) and (c) are redundant given the lines in the main figure. In particular, for Fig.2(c), it covers the experimental data for the in-plane dispersion of the charge-2 Weyl point at A.

Reply: We thank the referee for pointing out this issue. Now the insets in Figs. 2(b) and 2(c) have been removed, and the main text has been revised accordingly.

- What are the trivial surface states of Fig.3(d). Do the author understand why they appear? What is their extent in the surface Brillouin zone rather than as a function of ϕ ?

Figure: Isofrequency contours of the topologically trivial surface states (red dashed lines) evolved with frequencies, where the shadowed regions are bulk bands.

Reply: There is no definite physics origin for such topologically trivial surface states. In a common sense, surface states are *allowed* on a surface of a periodical structure since the surface momentum is a good quantum number. As shown in the above figure, the trivial surface states (red dashed lines) spread and evolve in the surface BZ. In particular, the isofrequency contours of the trivial surface states form close loops, much different from those of nontrivial ones that are open in geometry. In the Supplemental Materials (Fig. S3), we have provided a broadband isofrequency contours for the XZ surface, which display the evolution of topologically trivial and nontrivial states more clearly.

- I am slightly confused by the choice of the parameter ϕ . Currently it is taken in the counter-clockwise orientation. This gives states with a negative dispersion at Γ for example. But the topological charge of Γ is positive. It is purely a matter of convention but I find it confusing. I suggest to choose a clockwise convention for ϕ . Moreover, cylinders of the same size in Fig.3 and Fig. 4 would highly benefit the comparison between simulations and experimental data. Could the author choose a larger cylinder also in Fig.3?

Reply: We thank the referee for the kind suggestion. As suggested by the referee, we have chosen a clockwise convention for ϕ to replot the dispersions, and revised the main text accordingly.

As for the cylinder sizes involved in Fig. 3 and Fig. 4, we tend to retain the current data. First, in our experiments we use big cylinders (or loops in 2D surface BZ) to attain large band gaps, which benefits the identification of the topological surface

states. Since the momentum is discretized due to the finite-size effect, we extract the momentum grids near the circular loops to plot the experimental results. Specifically, in Fig. 4e we consider a loop of radius $0.43\pi/a$, other than that used in Fig. 4d, $0.5\pi/a$ (centered at $\bar{\Gamma}$). This enables more grids near the loop centered at \bar{H} , which also facilitates the demonstration of experimental dispersion. Accordingly, the numerical results for comparisons are simulated with circular loops of radii $0.5\pi/a$ and $0.43\pi/a$ for Fig. 4d and Fig. 4e, respectively. Second, in Fig. 3 we demonstrate all the numerical results with a *universal* cylinder radius, $0.2\pi/a$, which, in our opinion, enriches the numerical data involved in Fig. 4.

-In Fig.4, why the author overlay to the experimental data the simulated bulk states? I would possibly find these results more convincing if the experimentally measured bulk bands were displayed.

Reply: We thank the referee for the thoughtful comment. In our original manuscript, color shadows are intentionally used to sketch more clearly the boundaries of the simulated bulk states. To address this point, now the color shadows in Fig. 4 are made more transparent, and thus the measured bulk bands can be observed clearly. Similar revisions have also been made consistently in elsewhere.

- Two sources are used to excite the surface states. This, however, destroys the phase information for the measured signal. How do the authors circumvent this problem? Do they Fourier transform the signal of each exciter individually and then add the incoherently (with a square-sum)?

Reply: This is a very nice question. Technically, we summed the pressure fields excited by the two point sources directly and then performed Fourier transform to the total pressure field. This gives the amplitude information of the Fourier spectrum in our manuscript. As mentioned in our main text (lines 176-178), each point source mainly ignites one surface arc according to the known group velocity of surface states. As such, the influence of the initial phases of the point sources is weak to the amplitude of Fourier spectrum.

- Which smoothing function has been applied to the experimental data?

Reply: The figures are plotted by the software OriginPro 9.1 with “color contour” function, which smooth the discretized experimental data automatically.

- The authors could further elaborate on the open questions related to this work as the possible applications or possible transport studies in this platform under artificial fields.

Reply: We thank the referee for the constructive suggestion. Now we have added a sentence in the conclusion (lines 211-214) “Note that the isofrequency contours of topological surface states can be flexibly tailored through engineering the sample terminations²⁷, which may trigger new opportunities for controlling classical waves in a topological fashion, such as realizing reflectionless interface responses to sound²⁶⁻²⁹.”

The manuscript presents neat experimental data that substantiate the observation of a charge-2 Weyl point in an acoustic metamaterial. I believe that if the issues raised in this report will be addressed satisfactorily, the manuscript could meet the criteria of Nature Communications.

Reply: We thank the referee for the positive comment on this work. Based on the constructive suggestions from all the referees, our revised manuscript has been improved considerably. We wish the referee can suggest the publication of our work now.

Reviewer #3 (Remarks to the Author):

In their manuscript “Observation of quadratic Weyl points and double-helicoid arcs”, the authors present the first experimental realization of the quadratic Weyl points in a 3d acoustic metacrystal. The manuscript is well written, the introduction is enjoyable to read, and the results conclusively show the properties expected. Therefore, I recommend its publication in Nature Communications once the authors address the following comments.

Reply: We thank the referee for the high evaluation of our work, and also for his/her valuable suggestions and comments, which have been carefully addressed below.

1. My biggest concern comes from that the acoustic structure adopted in this work is very similar to the one in their previous work titled “Topological negative refraction of surface acoustic waves in a Weyl phononic crystal”, which was published in Nature. It looks like they study the same structure but focus on different properties, which reduces the novelty of this work. The authors should emphasize more standouts of it compared to their previous work.

Reply: We thank the referee for the constructive suggestion. To distinguish from our previous work more clearly, we have added two sentences in the introduction (lines 65-67), “Interestingly, this acoustic chiral crystal has been recently utilized to reveal novel topological negative refraction effect²⁷ at a frequency near the CWP. In this work, we focus on the topological physics around the QWP.”

2. In 2nd paragraph, 4th page, can the authors give the explicit expressions of coefficients A~F for their proposed structure? Is it possible to execute the calculation of Chern number (topological charge) using their effective Hamiltonians?

Reply: We thank the referee for the insightful question. In principle, we can calculate the coefficients numerically, and then calculate the topological charge using the effective Hamiltonian. Regrettably, after a careful inspection, we find that the provided Hamiltonian, which was derived straightforwardly from Refs. 19 and 30, cannot fully capture the feature of the quadratic dispersions around the DWP. Specifically, that Hamiltonian demands quadratic dispersions $|c|q^2$ and $-|c|q^2$ for the upper and lower bands touching at the DWP. However, this is not true for a general case, in which the quadratic curvatures of the two bands are independent, even hosting the same sign (e.g. the DWP at the point A, see Fig. 1d). Beyond doubt, it is very valuable to solve this problem, which is in fact what we have tried to do in the past month. However, we have not reached a satisfactory answer. Therefore, we decide to retain it as an open question and remove the effective Hamiltonian presented previously. In our opinion, this will not influence the main theme, novelty and impact of our *experimental* work.

3. Since topological charge at A is -2, I would expect surface states present near A are topologically protected, but why the authors claim surface states encircling around

Weyl node A are topologically trivial? How is the bulk-edge correspondence established in their case?

Figure: Some isofrequency contours of the surface states simulated for the k_x - k_z plane. The first three frequencies lie within the range of the surface states shown in Fig. 3d, and the next three frequencies locate at the overlapped frequency region between the lowest two bands. The surface states (red dashed lines) are topologically trivial since the isofrequency contours form close loops, much different from those topological ones, which are open in geometry.

Reply: The surface states plotted along the loop enclosing \bar{A} (Fig. 3d, main text) are topologically trivial. This can be conclusively deduced from the following two facts: (1) The frequency-dependent surface dispersion (red dashed line in Fig. 3d) is non-chiral; (2) The isofrequency contours of the surface states form close loops, instead of the open arcs that characterize nontrivial surface states. This can be seen from the above figures (red dashed lines).

In addition, no bulk-edge correspondence can be established for the loop around \bar{A} . As shown in Fig. 3d, there is no gap formed between the projected lowest two bulk bands (indicated by blue and pink), which is consistent with the two upward-sloping bulk dispersions along A-H (Fig. 1b).

To make the above points more clearly, we have made some revisions in our manuscript (lines 154-158), “For the loop enclosing \bar{A} (associated with the lower-frequency QWP at A), however, there is no gap between the projected lowest two bulk bands (Fig. 3d), consistent with the two upward-sloping bulk dispersions along A-H (Fig. 1b). As such, no topologically nontrivial surface states can be expected with bulk-boundary correspondence. (Note that the red dashed line corresponds to topologically trivial surface states since it is non-chiral along the loop.)”.

4. Some experimental details need to be supplied. For example, what is the frequency step in the frequency sweep (Fig. 2b,2c, Fig. 4d, 4e)? What are the quality factors of bulk modes and surface modes? In the experimental set up, what are the boundary conditions in z and x directions? and how are these boundary effects avoided or reduced on the Fourier spectrum?

Reply: We thank the referee for pointing out this issue. Now the aforementioned experimental details are supplied in our revised method (highlighted by yellow).

Specifically, the frequency step used is 32 Hz for each figure. The quality factors of the bulk modes and surface modes depend on the scanned spatial extension for each measurement. Specifically, the momentum resolutions are $\delta k_x = \frac{2\pi}{21a}$,

$\delta k_y = \frac{2\pi}{20\sqrt{3}a}$, and $\delta k_z = \frac{2\pi}{20a}$ along the three directions. During our bulk

measurements (Figs. 2b and 2c), all the six sample surfaces are exposed to air background, while to detect the surface states in Figs. 4d and 4e, an additional acrylic plate is tightly covered on the XZ surface. To reduce the boundary effects, the data near the boundaries of the measured plane are not taken into account when performing Fourier transform.

Reviewers' comments:

Reviewer #1 (Remarks to the Author):

The authors have addressed the reviewers' comments. I think the paper can be accepted for publication in NC.

Reviewer #2 (Remarks to the Author):

I am happy with the answer and changes provided by the authors to address my comments.

It is a pity for the difficulties encountered in fitting the free parameters of the effective Hamiltonian to the experimental data. I still think that such an analysis would greatly benefit the manuscript. Nevertheless, I agree with the authors that its absence does not diminish the value of the experimental data provided in the manuscript. My only remark is that without a Hamiltonian it is unclear where the dispersion of Fig.1d comes from. I would specify in the figure caption that these are schematic representations of the dispersions near the Weyl points.

I believe one last important issue remains to be addressed before I can recommend acceptance in Nature Communications. This concerns the use of two excitation sources for the surface state analysis. I understand the explanation provided by the authors to referee 1. To maximally excite different surface states, two different locations of the exciter are required.

This, however, does not justify the use of two sources but only two locations for two different data set. This is particularly relevant since the authors sum pressure field of the two sources rather than their squared sums (amplitudes). Such a procedure renders the phase information necessary to perform a Fourier transformation ill-defined. Since this is used to establish the topological charge of their degeneracies, this issue is of prominent relevance. I can agree with the authors that each source should (mainly) excite one surface state on theoretical grounds. Nevertheless, I do not consider a justification based on the expected theory sufficient. This analysis is one of the key ingredients of the manuscript and its validity should be established beyond any reasonable doubts.

I believe an experimental proof that each speaker excite only one arc is needed. Otherwise, the squared sum of the pressure field after the Fourier transform should be displayed and not the Fourier transform of the sum.

Reviewer #3 (Remarks to the Author):

The authors address most of my comments appropriately. However, based on their reply addressing my second comment, there is one aspect, which still needs further clarification. The information I get is, the effective Hamiltonians of their acoustic structure cannot be mapped to the ones in Refs. 19 and 30 (please correct me if I'm wrong). Since it's not sufficient to determine the topological charge by the chiral behavior of surface states along (the case of surface states near A is the counterexample), I would like to inquire authors providing more details of calculating the topological charge in their structure, what is the method they adopt to get the topological charges in Fig. 1b?

Response to Referees

Reviewer #1 (Remarks to the Author):

The authors have addressed the reviewers' comments. I think the paper can be accepted for publication in NC.

Reply: We thank the referee for the recommendation of publication in Nature Communications.

Reviewer #2 (Remarks to the Author):

I am happy with the answer and changes provided by the authors to address my comments.

Reply: We thank the referee for his/her great effort made to review our manuscript again.

It is a pity for the difficulties encountered in fitting the free parameters of the effective Hamiltonian to the experimental data. I still think that such an analysis would greatly benefit the manuscript. Nevertheless, I agree with the authors that its absence does not diminish the value of the experimental data provided in the manuscript. My only remark is that without a Hamiltonian it is unclear where the dispersion of Fig.1d comes from. I would specify in the figure caption that these are schematic representations of the dispersions near the Weyl points.

Reply: We thank the referee for pointing out this ambiguity. The dispersions in Fig. 1d are not schematic representations, but are simulated with the commercial software. For clarity, this has been pointed out explicitly in the figure caption (line 79) and the main text (line 94).

I believe one last important issue remains to be addressed before I can recommend acceptance in Nature Communications. This concerns the use of two excitation sources for the surface state analysis. I understand the explanation provided by the authors to referee 1. To maximally excite different surface states, two different locations of the exciter are required. This, however, does not justify the use of two sources but only two locations for two different data set. This is particularly relevant since the authors sum pressure field of the two sources rather than their squared sums (amplitudes). Such a procedure renders the phase information necessary to perform a Fourier transformation ill-defined. Since this is used to establish the topological charge of their degeneracies, this issue is of prominent relevance. I can agree with the authors that each source should (mainly) excite one surface state on theoretical grounds. Nevertheless, I do not consider a justification based on the expected theory sufficient. This analysis is one of the key ingredients of the manuscript and its validity should be established beyond any reasonable doubts.

I believe an experimental proof that each speakers excite only one arc is needed.

Otherwise, the squared sum of the pressure field after the Fourier transform should be displayed and not the Fourier transform of the sum.

Reply: We understand the concern of the referee. As suggested by the referee, we present an experimental proof in *Supplementary Materials* (see Fig. S3), which shows clearly that each speaker excites only one of the surface arcs.

Figure S3 | Surface arc excited by a single point source positioned at the corner of the sample. **a**, 2D Fourier transformation (color scale) of the surface field excited by a point source positioned at the bottom right corner of the sample at 8.64 kHz, compared with the numerical isofrequency contours of the topological surface states (red lines). As before, the color spheres highlight the projected Weyl nodes, and the shaded areas are the projected bulk bands. **b**, The same as **a**, but for the source positioned at the upper left corner. As expected, in each case the point source excites only one of the surface arcs according to the group velocities (labeled with arrows) of the surface states.

Reviewer #3 (Remarks to the Author):

The authors address most of my comments appropriately. However, based on their reply addressing my second comment, there is one aspect, which still needs further clarification. The information I get is, the effective Hamiltonians of their acoustic structure cannot be mapped to the ones in Refs. 19 and 30 (please correct me if I'm wrong). Since it's not sufficient to determine the topological charge by the chiral behavior of surface states along (the case of surface states near A is the counterexample), I would like to inquire authors providing more details of calculating the topological charge in their structure, what is the method they adopt to get the topological charges in Fig. 1b?

Reply: We thank the referee for carefully reviewing our revised manuscript. We also thank the referee for the constructive suggestion. Now we have added a sentence in the revised manuscript (see lines 96-98) to point out the method adopt to get the topological charges in Fig. 1b, “The topological charges of the CWPs and QWPs, obtained by analyzing the eigenvalues of the threefold screw symmetry at their high-symmetry momenta^{19,30}, are highlighted in Figs. 1b and 1c.”